# Fair Surveillance Assignment Problem

## ABSTRACT

Monitoring a specific set of locations serves multiple purposes in practice, such as infrastructure inspection and safety surveillance. We study a generalization of the surveillance problem, where the monitoring area, represented by a graph, is divided and assigned to a set of agents with customized cost functions. In this paper, each agent's patrolling cost towards receiving a subgraph is quantified by the weight of the minimum vertex cover therein, and our objective is to design algorithms to compute fair assignments of the surveillance tasks. The fairness is assessed using maximin share (MMS) fairness proposed by Budish [J. Political Econ., 2011]. Our main result is an algorithm which ensures a $\frac{5+\sqrt{17}}{2}(\approx 4.562)$-approximate MMS allocation for any number of agents with arbitrary vertex weights. We then prove that no algorithm can be better than 2-approximate MMS fair. For scenarios involving no more than four agents, we further improve the approximation ratio to 2, which is thus the optimal achievable ratio.

## KEYWORDS

Surveillance, Fair Division, Indivisible Chores, Vertex Cover

**ACM Reference Format:**
Anonymous Author(s). 2024. Fair Surveillance Assignment Problem. In *Proceedings of The Web Conference 2024 (WWW'24).* ACM, New York, NY, USA, 9 pages. https://doi.org/XXXXXXX.XXXXXXX

## 1 INTRODUCTION

A wide range of patrolling and surveillance models have been investigated to ensure public safety, a pivotal aspect of everyday life. Given the daunting task of monitoring vast areas, a key challenge lies in effectively dividing the monitored area among multiple patrollers. There is a rich body of literature on designing optimal patrolling assignments, considering the capabilities of the patrollers. The primary objective typically revolves around global optimization, aiming to enhance overall performance or bolster the weakest link [9, 30, 38]. Importantly, when human agents are involved, it becomes imperative to ensure fairness in assignments, particularly for long-term monitoring undertakings, as it promotes sustainability. Thus, in this paper, we introduce a novel surveillance model from the perspective of the patrollers. Our objective is to develop individually fair patrolling assignments while guaranteeing comprehensive coverage of the entire area.

We cast the above fair surveillance assignment task as a graph partition $G = (V, E)$ problem, where the edges $E$ represent the

streets on which the residents locate and the vertices $V$ are the intersections of these streets. Our task is to allocate the streets $E$ to a set of agents $\mathcal{N}$ for surveillance. Each agent $a_i$, upon receiving a set of streets $S \subseteq E$, would set up monitors on the vertices in a vertex cover of $G[S]$, the induced subgraph of $S$ on $G$, so that all the streets are safely monitored. We consider a general setting when different agents may have perspectives on the difficulties of monitoring the vertices, due to their personal familiarity of the neighborhoods and the number of incident streets. Formally, each agent $a_i$ is associated with a vertex weight $w_{i,v} > 0$ for each $v \in V$, which represents the difficulty of setting up a monitor at $v$ surveilling the incident streets. Accordingly, $a_i$'s cost function $c_i : 2^E \to \mathbb{R}^+$ can be measured by vertex covers, i.e., for any $S \subseteq E$, $c_i(S)$ equals the minimum weight of a vertex cover on $G[S]$. Consider a simple example of the graph shown in Figure 1. Suppose the agent has vertex weight as follows: $w_{i,a} = w_{i,c} = 1$ and $w_{i,b} = w_{i,d} = 10$. Then $c_i(\{a,b\}, \{b,c\}) = w_{i,a} + w_{i,c} = 2$ and $c_i(\{a,b\}, \{a,d\}) = w_{i,a} = 1$.

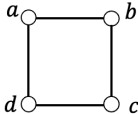

**Figure 1: A simple example of the cost functions.**

We adopt Budish's maximin share (MMS) definition [8] to evaluate the fairness of an allocation. Budish's original definition was proposed for allocating goods where agents want to receive items with higher value, but it naturally carries over chores. Intuitively, we define a best worst-case cost, called maximin share (MMS), for each agent: the agent partitions the edges $E$ into $n = |\mathcal{N}|$ bundles but can only receive the largest bundle (i.e., the bundle with largest cost), and her MMS is the minimum cost of the largest bundles over all possible partitions. An allocation is regarded as MMS fair, if every agent's cost of her obtained edges is no greater than her MMS. It can be verified that the agents' cost functions in our model are subadditive, and thus our problems falls under the umbrella of [27] which studies approximate MMS allocations of indivisible chores under general subadditive functions. They proved that an arbitrary profile of subadditive functions may not admit a better than $n$-approximate MMS allocation, and in a sharp contrast, we prove in the current work that, if the agents' costs are measured by vertex covers, a constant approximate MMS allocation is guaranteed to exist. We summarize our main results in the following subsection.

### 1.1 Our Results

In this paper, we propose the *Fair Surveillance Assignment Problem* (FSAP), where the edges of a given graph $G = (V, E)$ need to be assigned to a set of $n$ agents for surveillance. The agents' cost functions depend on the weight of the vertex covers of their received

subgraph using their own weight metric. We focus on the MMS fairness of the allocations.

For the general FSAP, we first prove that no algorithm can guarantee approximation ratio of MMS better than 2. This hardness result holds for arbitrary number of agents $n \geq 2$.

**Main Result 1.** For any $n \geq 2$ agents, there is a FSAP instance which does not admit a better than 2-approximate MMS allocation.

On the positive side, we show that we can always compute a $\frac{5+\sqrt{17}}{2} (\approx 4.562)$-approximate MMS allocation for any FSAP instance with $n \geq 2$ agents, which is regarded as the main technical contribution of this work. For each agent $a_i$, our algorithm first identifies the vertices $M_i$ used in her MMS partition, and regards them as valid vertices to $a_i$. The algorithm ensures that every agent only receives the edges that can be covered by her valid vertices. The algorithm runs in rounds; in each round, one of the agents receives a bundle that satisfies the desired approximation ratio and exists the algorithm. The key ingredient of the algorithm is the *pivotal vertices* of each round which, informally, consists of the vertices that are valid to at least half of the remaining agents. The pivotal vertices are allocated to the agents one by one in a greedy way while ensuring the approximation ratio. The first time when the approximation ratio is broken by some agent, her allocation is fixed and the vertices allocated to the other agents are returned to the algorithm. The algorithm then updates the graph and the pivotal vertices, and moves to the next iteration. We prove the correctness of the algorithm via a cumulative counting approach: the agent who leaves the algorithm in each round takes away edges with sufficiently large weight to all the remaining agents. Our algorithm runs in polynomial time if the agents' valid vertices are given as oracle.

**Main Result 2.** For any FSAP instance with $n \geq 2$ agents, there always exists a 4.562-approximate MMS allocation.

Finally, we show that for the scenario with no more than four agents, we can improve the approximation to 2. Combining with Main Result 1, this is the best possible approximation ratio. The proof depends on an involved case analysis, and it is still now clear if these techniques can be extended to improve the approximations for more than four agents, which is left for future research.

**Main Result 3.** For any FSAP instance with $2 \leq n \leq 4$ agents, there always exists a 2-approximate MMS allocation.

## 1.2 Related Work

More generally, extensive research has been conducted in operations research, computer science and Web economics regarding the partitioning of graphs into balanced subgraphs [9, 30]. Various objectives are commonly used to evaluate the balance of a partitioning. Among these objectives, the max-min (or min-max) objectives are particularly prominent, aiming to maximize (or minimize) the total weight of the minimum (or maximum) part. Our project is closely related to the vehicle routing problem (VRP) [25], which extends the well-known traveling salesperson problem (TSP). The VRP involves finding optimal routes for a fleet of vehicles to visit a set of customers. There exist several popular variants of the VRP, such as the heterogeneous vehicle routing problem [32, 33, 38]. The

graph partitioning problem encompasses various other combinatorial structures. For example, the min-max tree cover (also known as the nurse station location) problem focuses on using trees to cover an edge-weighted graph while minimizing the size of the largest tree [15, 24, 37]. This problem is a part of the broader graph covering problem, where a given graph is covered using a set of pairwise disjoint subgraphs known as templates. Examples of templates include paths [4, 16], cycles [28, 35], network flows [7], and matchings [23, 26].

Our model differs from the aforementioned research in several perspectives. Mostly importantly, we consider customized cost functions, where different agents possesses different weights on the vertices, and the fairness is measured by the MMS. Moreover, most surveillance models [1, 2] assume the costs are measured by TSP, while we focus on the minimum vertex cover.

Another line of research that is closely related to our work is fair division. The "fairness" of an allocation can be interpreted in several ways [5, 12], where different people may have different opinions. Two predominant classes of fairness criteria are the envy-based or share-based. An envy-based criterion evaluates an agent's demand for fairness on an allocation through pair comparison between this agent and each of her peers (e.g., envy-freeness) [17, 36], and a share-based criterion evaluates that against the agent's due share (e.g., proportionality) [34]. When the resources are *indivisible* items, the problem turns out to be inherently more challenging since the ideal fairness notions cannot be always satisfied. A typical remedy is to employ appropriate *relaxations* of envy-freeness and proportionality, originating in the works of [8, 10, 19, 29], which are geared to escape such adverse examples. Since then, works on this topic have flourished, centered around fundamental questions about the existence, approximations, and the efficient computation of allocations satisfying these or other related fairness criteria; we refer to recent surveys [3] for a more comprehensive introduction. Among the relaxations that were introduced in this literature, *envy-freeness up to one or any good* (EF1, EFX) [11, 31, 39] and *maximin share fairness* (MMS) are among the most well-accepted and studied [14, 18, 22]. It is proved in [6] that an EF1 allocation always exists for any monotone combinatorial cost functions, which implies the existence of an EF1 allocation in our setting.

There are some recent works that combines fair division and graphs, see, e.g., [13, 20, 21], surveillance models have not considered. While it is proved in [27] that for general submodular cost functions, no better than $n$-approximate MMS allocation can be guaranteed, our result shows that vertex cover costs serve an exception which allows interesting and non-trivial approximations.

## 2 PRELIMINARIES

### 2.1 Fair Surveillance Assignment Problem

We firstly introduce the definitions of our model. A *Fair Surveillance Assignment Problem* (FSAP) instance is denoted by $\mathcal{I} = (\mathcal{N}, G, \mathbf{c})$, where $G = (V, E)$ is a graph whose edges $E = \{e_1, \ldots, e_m\}$ are to be allocated to a set $\mathcal{N} = \{a_1, \ldots, a_n\}$ of agents. Let $n = |\mathcal{N}|$ and $m = |E|$ be the numbers of agents and edges, respectively. Each agent $a_i \in \mathcal{N}$ has weight $w_{i,v} \in \mathbb{R}^+$ for each vertex $v \in V$, and $\mathbf{w}_i = (w_{i,v})_{v \in V}$. Agent $a_i$'s cost function $c_i : 2^E \to \mathbb{R}^+$ is decided by solving a vertex cover problem, i.e., $c_i(S)$ equals the minimum

weight of a vertex cover of $G[S]$ for every $S \subseteq E$, where a vertex cover is a set of vertices $T$ in $G[S]$ so that every edge in $S$ has at least one endpoint in $T$. Supposing $G[S] = (V', S)$ and $T \subseteq V'$ is a minimum weighted vertex cover of $G[S]$, we also denote $c_i(S)$ by $c_i(G[S])$ or $c_i(T) = \sum_{v \in T} w_{i,v}$, when there is no ambiguity. Denote by $\mathbf{c} = (c_1, \ldots, c_n)$. Without loss of generality, we assume there is no isolated vertices (i.e., with degree 0) in $G$.

It is not hard to see that the cost functions are (weakly) monotonically increasing and subadditive. To see the subadditivity, for any agent $a_i$ and any $S_1, S_2 \subseteq E$, suppose the minimum weighted vertex covers of $G[S_1]$ and $G[S_2]$ are $T_1$ and $T_2$. Then $T_1 \cup T_2$ is also a vertex cover of $G[S_1 \cup S_2]$, and

$$c_i(S_1 \cup S_2) \leq \sum_{v \in T_1 \cup T_2} w_{i,v} \leq \sum_{v \in T_1} w_{i,v} + \sum_{v \in T_2} w_{i,v} = c_i(T_1) + c_i(T_2).$$

An allocation of instance $\mathcal{I} = (\mathcal{N}, G = (V, E), \mathbf{c})$ is an $n$-partition of $E$, denoted by $\mathbf{A} = (A_1, \ldots, A_n)$, where $A_i$ is the allocation to agent $a_i$ for all $a_i \in \mathcal{N}$, $\bigcup_{a_i \in \mathcal{N}} A_i = E$ and $A_i \cap A_j = \emptyset$ for all $a_i \neq a_j$. Note that when two adjacent edges $e_1 = (v_1, v_2)$ and $e_2 = (v_1, v_3)$ sharing the same vertex $v_1$ are allocated to two different agents, say $a_i$ and $a_j$, then $v_1$ appears in both agents' induced subgraphs and thus both of them may use $v_1$ in their vertex covers. In this work, for simplicity, we assume the weight of $v_1$ is not changed to the two agents even if it is incident to fewer number of edges in their own subgraphs (although it may potentially reduce difficulty of monitoring at this vertex). Denoted by $\Pi_n(M)$ the set of all $n$-partitions of $E$.

## 2.2 MMS Fairness

Next, we introduce the definition of maximin share fairness [8]. For an agent $a_i \in \mathcal{N}$, her maximin share (MMS) is defined as

$$\mu_i^n = \min_{(X_1, \ldots, X_n) \in \Pi_n(E)} \max_{j \in [n]} c_i(X_j).$$

When $n$ is clear from the context, we also write $\mu_i$ for short. An $n$-partition $(X_1, \ldots, X_n)$ of $E$ is called an MMS defining partition of agent $a_i$ if

$$c_i(X_j) \leq \mu_i, \text{ for all } j = 1, \ldots, n.$$

DEFINITION 1 (MAXIMIN SHARE ALLOCATION). *An allocation* $\mathbf{A} = (A_1, \ldots, A_n)$ *is called $\alpha$-approximate MMS ($\alpha$-MMS) fair if*

$$c_i(A_i) \leq \alpha \cdot \mu_i \text{ for all } a_i \in \mathcal{N}.$$

*The allocation is called MMS fair if $\alpha = 1$.*

Since the cost functions are subadditive in our model, by [27], there always exists an allocation that is $\min\{n, \lceil \log m \rceil\}$-MMS allocation. In this paper, we improve this result to constant approximations. Before introducing our main results, we first prove that no algorithm can perform better than 2-MMS.

THEOREM 1. *For any $n \geq 2$, there is a FSAP instance with $n$ agents which does not admit a better than 2-MMS allocation.*

PROOF. Given any $n \geq 2$, we construct a complete $n$-by-$n$ bipartite graph $G = (L \cup R, E)$ as Fig. 2 shows, where $L = \{v_1, \ldots, v_n\}$, $R = \{u_1, \ldots, u_n\}$ and $E = \{(v_i, u_j) \mid \text{for all } v_i \in L \text{ and } u_j \in R\}$. For any $v_i \in L$ and $u_j \in R$, denote by $E_{v_i} = \{(v_i, u_l) \mid \text{for all } u_l \in R\}$

and $E_{u_j} = \{(v_l, u_j) \mid \text{for all } v_l \in L\}$ the sets of all edges incident with $v_i$ and $u_j$, respectively. Note that both $(E_{v_1}, \ldots, E_{v_n})$ and $(E_{u_1}, \ldots, E_{u_n})$ are partitions of $E$.

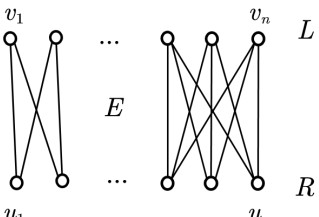

**Figure 2: The bipartite graph $G$.**

Next, we design the cost functions of $n$ agents $\mathcal{N} = \{a_1, \ldots, a_n\}$. For any agent $a_i$, where $i = 1, \ldots, n-1$, let $w_{i,j} = 1$ for any $v_j \in L$ and $w_{i,l} = n^2$ for any $u_l \in R$, and her cost function $c_i(\cdot)$ is defined accordingly. Then $\mu_i = 1$, which is obtained by partitioning the edges $E$ as $(E_{v_1}, \ldots, E_{v_n})$ since $c_i(E_{v_j}) = 1$ for all $j = 1, \ldots, n$. For agent $a_n$, let $w_{n,j} = 1$ for any $u_j \in R$ and $w_{n,l} = n^2$ for any $v_l \in L$, and her cost function $c_n(\cdot)$ is defined accordingly. Then $\mu_n = 1$, which is obtained by partitioning the edges $E$ as $(E_{u_1}, \ldots, E_{u_n})$ since $c_n(E_{u_j}) = 1$ for all $j = 1, \ldots, n$.

For the sake of contradiction, suppose instance $\mathcal{I} = (\mathcal{N}, G, \mathbf{c})$ admits a better than 2-MMS allocation $\mathbf{A} = (A_1, \ldots, A_n)$. By the design of the instance, it must be that $c_i(A_i) = 1$ for all $a_i \in \mathcal{N}$. Thus, for $1 \leq i \leq n-1$, $A_i$ cannot contain edges from more than one $E_{v_l}$, i.e., $A_i \subseteq E_{v_l}$ for some $v_l \in L$. By the pigeon-hole principle, there exists $v_{l^*} \in L$ such that $E_{v_{l^*}} \cap (A_1 \cup \cdots \cup A_{n-1}) = \emptyset$, which implies $E_{v_{l^*}} \subseteq A_n$. However, since $E_{v_{l^*}} \cap E_{u_j} \neq \emptyset$ for all $u_j \in R$, $c_n(A_n) \geq c_n(E_{v_{l^*}}) = \sum_{u_j \in R} w_{n,j} = n$, which contradicts $c_n(A_n) = 1$ and completes the proof of Theorem 1. □

# 3 THE MAIN ALGORITHM

## 3.1 More Notations

We first introduce more notations which will be used in the description of the algorithm. One of the difficulties in designing effective fair algorithms to allocate the edges is that the agents' costs actually depend on vertices of the induced subgraph and moreover only the weight of a subset of these vertices gives the cost of the agent. Therefore, the first step of our algorithm is to identify some critical vertices which can be used to approximate the agents' costs.

Consider an arbitrary agent $a_i \in \mathcal{N}$. Note that there may be multiple partitions of $E$ that gives the MMS value of agent $a_i$. We arbitrarily fix one of such partitions and denote it by $(E_{i,1}, \ldots, E_{i,n})$, where $c_i(E_{i,j}) \leq \mu_i$ for $j = 1, \ldots, n$ and there exists $j^*$ such that $c_i(E_{i,j^*}) = \mu_i$. Suppose $G[E_{i,j}] = (V_{i,j}, E_{i,j})$ and $M_{i,j} \subseteq V_{i,j}$ is one minimum weighted vertex cover of $G[E_{i,j}]$. Again, $M_{i,j}$ may not be the unique minimum vertex cover, in which case we arbitrarily fix one of them. Denote $M_i = M_{i,1} \cup \cdots \cup M_{i,n}$, which are called *valid vertices* for agent $a_i$. Let $\overline{M_i} = V \setminus M_i$ be the set of invalid vertices for agent $a_i$.

Since every $M_{i,j}$ covers $G[E_{i,j}]$, it is easy to see that $M_i$ covers the original graph $G$. However, $M_i$ may not be a minimum vertex cover. Consider a star graph with one internal vertex and $n$ leaves,

where the internal vertex weighs 2 to $a_i$ and each leaf weighs 1. Then the weight of a minimum vertex cover is 2 (the internal vertex suffices), while $\mu_i = 1$ by partitioning all edges into isolated edges and thus all leaves form the valid vertices $M_i$ for $a_i$. In general, we have the following property of the valid vertices.

**Lemma 1.** *For any $a_i \in \mathcal{N}$ and $S \subseteq E$ with $G[S] = (V', S), V' \cap M_i$ is a vertex cover of $G[S]$ and thus*

$$c_i(S) \leq \sum_{v \in V' \cap M_i} w_{i,v}.$$

**Proof.** It suffice to see $V' \cap M_i$ covers $G[S]$. This is straightforward, as otherwise, there is an edge $e \in S$ that is not covered by $V' \cap M_i$ and thus not by any vertex in $M_i$, which contradicts $M_i$ covers $G$. $\square$

Next, it is easy to observe the following property of MMS.

**Lemma 2.** *For any $a_i \in \mathcal{N}$,*

$$\mu_i \geq \max \left\{ \max_{v \in M_i} w_{i,v}, \frac{1}{n} \cdot \sum_{v \in M_i} w_{i,v} \right\}.$$

**Proof.** Suppose $(E_{i,1}, \ldots, E_{i,n})$ is the MMS-defining partition that is used to define the valid vertices $M_i = M_{i,1} \cup \cdots \cup M_{i,n}$ for agent $a_i$. To see $\mu_i \geq \max_{v \in M_i} w_{i,v}$ and $\mu_i \geq \frac{1}{n} \cdot \sum_{v \in M_i} w_{i,v}$, it suffices to recall the definition of $\mu_i$, i.e.,

$$\mu_i = \max_{j=1,\ldots,n} c_i(E_{i,j}) = \max_{j=1,\ldots,n} \sum_{v \in M_{i,j}} w_{i,j}.$$

$\square$

*Remark.* We remark that although the selection of $M_i$ in the previous discussion is not unique, any of them suffices to prove the approximation ratio of our algorithm.

By Lemma 1, we can use the valid vertices to upper bound the cost of any set of edges allocated to agent $a_i$. Furthermore, instead of allocation edges, we can also regard the problem as allocating the vertices in $M_i$ and thus if $a_i$ receives some vertices in $M_i$, she also receives all their incident edges. However, the difficulty is that different agents have different valid vertices. To overcome this difficulty, we introduce the definition of *pivotal vertices*, which will be formally defined in the algorithm.

The use of pivotal vertices relies on the following lemma. Given a graph $G = (V, E)$ and $k$ vertex covers $\mathbf{C} = \{C_1, \ldots, C_k\}$ of $G$. For all $v \in V$, denote by $N_v = \{i \in [k] \mid v \in C_i\}$ the vertex covers that use $v$. Let

$$P = \{v \in V \mid |N_v| \geq \frac{k}{2}\},$$

be the vertices that for which at least half of $\mathbf{C}$ use them in the vertex covers. Then we have the following property.

**Lemma 3.** *$P$ is a vertex cover of $G$.*

**Proof.** We prove by contradiction. Suppose there is an edge $(v_1, v_2) \in E$ that is not covered by $P$. By the definition of $P$, $|N_{v_1}| < \frac{k}{2}$ and $|N_{v_2}| < \frac{k}{2}$, and thus $|N_{v_1} \cup N_{v_2}| < k$. Therefore, there is at least one $i^* \in [k] \setminus (N_{v_1} \cup N_{v_2})$ such that $v_1 \notin C_{i^*}$ and $v_2 \notin C_{i^*}$, which is a contradiction with $C_{i^*}$ being a vertex cover of $G$. $\square$

## 3.2 The Algorithm

In this section, we formally describe our algorithm and prove its approximation.

**Theorem 2.** *For any SAP instance $\mathcal{I} = (\mathcal{N}, G, \mathbf{c})$ with $n \geq 2$ agents, Algorithm 1 computes a $\frac{5+\sqrt{17}}{2} (\approx 4.562)$-MMS allocation.*

We first note that without loss of generality, we can always focus on normalized instances where $\mu_i = 1$ for all $a_i \in \mathcal{N}$. On way to achieve this is to divide $w_{i,v}$ by $\mu_i$ for all $v \in V$, which only affects the weights of vertex covers but not the selections of them.

The algorithm is shown in Algorithm 1. Basically, the algorithm runs in rounds with a parameter $\alpha$ which is defined in the proof of Theorem 2. In each round, it identifies a set of pivotal vertices $P$ where each vertex is valid to at least half of all remaining agents. Note that $P$ is updated in every round. Then we process the vertices in $P$ in an arbitrary order. For each pivotal vertex $v \in P$, among all the agents for whom $v$ is valid, we allocate it to the one with smallest weight. As soon as one agent $i^*$'s weight exceeds $\alpha$, we stop allocating the vertices. Agent $i^*$'s allocation is finalized to be the edges that are covered by pivotal vertices allocated to her in the current round. Then the algorithm excludes $i^*$, returns all the vertices allocated to the other agents, and moves to the next round. If all vertices in $P$ are allocated without making any agent's total weight exceed $\alpha$, then each edge is allocated to (arbitrary) one of the agents whose obtained pivotal vertex covers it.

**Proof of Theorem 2.** We first see that Algorithm 1 is well-defined. Let $M_i$ be the valid vertices $M_i$ of every agent $a_i$, and by Lemma 1, $M_i$ is a vertex cover of the input graph $G$ and thus also a vertex cover of any subgraph of $G$. Through the run of the algorithm, $G = (V, E)$ and $\mathcal{N}$ represent the remaining subgraph and the remaining agents. We call each iteration of the **while** loop at Step 2 a *round* of the algorithm. By Lemma 3, the set of pivotal vertices $P$ computed in Step 4 is nonempty and covers $G$. During the execution of the **for** loop in Step 7, for every vertex $v \in P$, by the definition of $P$, it is valid to at least one agent. Then at the end of this **for** loop either one agent $a_{i^*}$'s weight exceeds $\alpha$ or all vertices in $P$ are allocated. For the former case, agent $a_{i^*}$ obtains the edges covered by her allocated pivotal vertices, $G$ and $\mathcal{N}$ are updated, and the algorithm moves to the next round. For the later case, since $P$ is a vertex cover, every edge must be incident to at least one vertex in $P$ and thus is allocated to some agent in the **for** loop at Step 17, and the algorithm ends here.

Next, we prove the approximation ratio of the algorithm by setting $\alpha = \frac{3+\sqrt{17}}{2} \approx 3.562$. Recall that $\mu_i = 1$ for all $a_i \in \mathcal{N}$. By the design of the algorithm, every agent $a_i$ can only receive a bundle (1) from Step 13 or (2) from the **for** loop at Step 17. For case (1), there is one round such that all the vertices in $P$ are allocated, which means that the **for** loop at Step 7 is not broken by any agent. Thus,

$$c_i(A_i) \leq \sum_{v \in B_i} w_{i,v} \leq \alpha = \alpha \cdot \mu_i,$$

where the first inequality is by Lemma 1, recalling that $B_i$ is a vertex cover of $G[A_i]$. For case (2), agent $a_i$ breaks the **for** loop at Step 7, which is the first time when her weight for $B_i$ exceeds $\alpha$. Suppose

**Algorithm 1** Approximation Algorithm for General Instances.

**Input:** A normalized instance $I = (N, G = (V, E), \mathbf{c})$ with valid vertices $M_i$ for all $a_i \in N$, and parameter $\alpha > 1$.

**Output:** An allocation $\mathbf{A} = (A_1, \ldots, A_n)$.

1: Initialize $A_i = \emptyset$ for all $a_i \in N$.
2: **while** $N \neq \emptyset$ and $E \neq \emptyset$ **do**
3:     Set $N_v \leftarrow \{a_i \in N \mid v \in M_i\}$ for all $v \in V$.
4:     Set pivotal vertices

$$P \leftarrow \{v \in V \mid |N_v| \geq \frac{|N|}{2}\}.$$

5:     Set $B_i \leftarrow \emptyset$ for all $a_i \in N$.
6:     Let $i^* \leftarrow \perp$.
7:     **for** $v \in P$ **do**
8:         Let $a_i$ be an agent for whom $v$ is valid and the weight on $v$ is smallest, i.e.

$$a_i \in \arg\min_{a_j \in N_v} w_{j,v}.$$

        Set $B_i \leftarrow B_i \cup \{v\}$.
9:         If $\sum_{v \in B_{i^*}} w_{i^*,v} \geq \alpha$, set $i^* \leftarrow i$ and break the **for** loop.
10:     **end for**
11:     **if** $i^* \neq \perp$ **then**
12:         Let $S \subseteq E$ be the set of edges covered by $B_{i^*}$.
13:         Set $A_{i^*} \leftarrow S$.
14:         Set $N \leftarrow N \setminus \{a_{i^*}\}, E \leftarrow E \setminus S$.
15:         Set $G \leftarrow G[E]$ and disregard all isolated vertices.
16:     **else**
17:         **for** $e = (v_1, v_2) \in E$ **do**
18:             Choose an arbitrary agent $a_i$ such that $B_i \cap \{v_1, v_2\} \neq \emptyset$.
19:             Set $A_i \leftarrow A_i \cup \{e\}$.
20:         **end for**
21:         **return** $\mathbf{A} = (A_1, \ldots, A_n)$.
22:     **end if**
23: **end while**
24: **return** $\mathbf{A} = (A_1, \ldots, A_n)$.

$v^*$ is the last vertex added to $B_i$, then

$$c_i(A_i) \leq \sum_{v \in B_i \setminus \{v^*\}} w_{i,v} + w_{i,v^*} < \alpha + w_{i,v^*} \leq (\alpha + 1) \cdot \mu_i, \quad (1)$$

where the first inequality is by Lemma 1 and the last inequality is by Lemma 2.

It remains to show all the edges can be allocated at the end of the algorithm, by setting $\alpha = \frac{3+\sqrt{17}}{2} \approx 3.562$. We prove by contradiction, and suppose there are edges left unallocated. Thus all agents obtain their allocations from Step 13 and the algorithm must run $n$ rounds of **while** loop at Step 2, where $n$ is the initial number of agents. Suppose the sets of the pivotal vertices used in each rounds are $P_1, \ldots, P_n$, and rename the agents so that agent $a_k$ exists from the algorithm in round $k = 1, \ldots, n$. We first define the following quantity $\Delta$:

$$\Delta = \sum_{k=1}^{n} \sum_{j=k}^{n} \sum_{v \in B_k \cap M_j} w_{k,v},$$

where $\sum_{v \in B_k \cap M_j} w_{k,v}$ means when agent $a_k$ exists, the total weight she takes way from every remaining agent $a_j$'s valid vertices, in $a_k$'s perspective.

On the one hand, we note that for each vertex $v \in B_k \subseteq P_k$, there are at least $\frac{n-k+1}{2}$ agents for whom $v$ is valid, which means $v$ appears in $B_k \cap M_j$ for at least $\frac{n-k+1}{2}$ terms. Thus, by rearranging the weights, we have

$$\sum_{j=k}^{n} \sum_{v \in B_k \cap M_j} w_{k,v} \geq \frac{n-k+1}{2} \sum_{v \in B_k} w_{k,v} \geq \frac{n-k+1}{2} \cdot \alpha.$$

Therefore,

$$\Delta \geq \sum_{k=1}^{n} \frac{n-k+1}{2} \cdot \alpha \geq \frac{\alpha}{4} n(n+1). \quad (2)$$

On the other hand, exchanging the order of summations gives

$$\Delta = \sum_{k=1}^{n} \sum_{j=k}^{n} \sum_{v \in B_k \cap M_j} w_{k,v} = \sum_{j=1}^{n} \sum_{k=1}^{j} \sum_{v \in B_k \cap M_j} w_{k,v}$$

$$= \sum_{j=1}^{\lfloor n/(\alpha+1) \rfloor} \sum_{k=1}^{j} \sum_{v \in B_k \cap M_j} w_{k,v} + \sum_{j=\lfloor n/(\alpha+1) \rfloor+1}^{n} \sum_{k=1}^{j} \sum_{v \in B_k \cap M_j} w_{k,v},$$

where the left and the right terms are denoted by $\Delta_1$ and $\Delta_2$, respectively. For $\Delta_1$, we have

$$\Delta_1 = \sum_{j=1}^{\lfloor n/(\alpha+1) \rfloor} \sum_{k=1}^{j} \sum_{v \in B_k \cap M_j} w_{k,v} < \sum_{j=1}^{\lfloor n/(\alpha+1) \rfloor} \sum_{k=1}^{j} (\alpha + 1)$$

$$= \sum_{j=1}^{\lfloor n/(\alpha+1) \rfloor} j \cdot (\alpha + 1) \leq \frac{\alpha+1}{2} \frac{n}{\alpha} \left( \frac{n}{\alpha} + 1 \right),$$

where the first inequality is because $\sum_{v \in B_k \cap M_j} w_{k,v} < \alpha+1$, similar as Inequality 1. For $\Delta_2$, we have

$$\Delta_2 = \sum_{j=\lfloor n/(\alpha+1) \rfloor+1}^{n} \sum_{k=1}^{j} \sum_{v \in B_k \cap M_j} w_{k,v}$$

$$\leq \sum_{j=\lfloor n/(\alpha+1) \rfloor+1}^{n} \sum_{k=1}^{j} \sum_{v \in B_k \cap M_j} w_{j,v}$$

$$\leq \sum_{j=\lfloor n/(\alpha+1) \rfloor+1}^{n} \sum_{v \in M_j} w_{j,v}$$

$$\leq \sum_{j=\lfloor n/(\alpha+1) \rfloor+1}^{n} n = \left( n - \lfloor \frac{n}{\alpha+1} \rfloor \right) \cdot n,$$

where the first inequality is because $a_k$ has the smallest weight among the agents for whom $v \in B_k$ is valid and the last inequality is because $\mu_j = 1$. Combining the above two inequalities, we have

$$\Delta < n^2 (1 - \frac{1}{2(\alpha+1)}) + \frac{3n}{2} \quad (3)$$

However, when $\alpha = \frac{3+\sqrt{17}}{2}$, Inequality 2 contradicts Inequality 3. Therefore, it must be that there is round the algorithm reaches the **for** loop in Step 17, and thus all the edges are allocated. □

*Remark.* Note that given the valid vertices $M_i$ for every agent as an oracle, Algorithm 1 runs in polynomial time. There are at most $n$ rounds of the outer **while** loop. Within each round, computing the pivotal vertices (Step 4) and finding the largest weight on these vertices (the **for** loop at Step 7) need $O(n|V|)$ time. The **for** loop at Step 17 is only executed once, which requires $O(|E|)$ time. However, the computation of $M_i$ is NP-hard, and it is still not clear how to efficiently approximate them, which is left as an open problem of this paper.

## 4 IMPROVED APPROXIMATIONS FOR SMALL NUMBER OF AGENTS

In this section, we prove that when the instance contains no more than four agents, there is always a 2-MMS allocation. Combining with Theorem 1, it is the optimal approximation ratio for these cases.

THEOREM 3. *For any FSAP instance* $\mathcal{I} = (\mathcal{N}, G = (V, E), \mathbf{c})$ *with* $|\mathcal{N}| \leq 4$, *there exists a 2-MMS allocation.*

It is observed in [27] that for subadditive cost functions, an arbitrary allocation, including the one that allocates all items to a single agent, is $n$-MMS. Thus, when $n = 2$, we can simply allocate all edges to an arbitrary agent, which gives a 2-MMS allocation.

When $n > 2$, the problem becomes tricky. We show how to find a 2-MMS allocation for $n = 2, 3$ in Subsections 4.1 and 4.2 (see Lemmas 4 and 5 respectively). It is not clear if a 2-MMS allocation is guaranteed to exist or not, which is left for future study.

### 4.1 Three Agents

In this subsection, we show how to find a 2-MMS allocation for any FSAP instance $\mathcal{I} = (\mathcal{N}, G = (V, E), \mathbf{c})$ with 3 agents, i.e., $\mathcal{N} = \{a_1, a_2, a_3\}$. Arbitrarily fix two agents $a_i$ and $a_j$, we partition the vertices $V$ into four parts: (1) $X_{i,j} = M_i \cap M_j$ which contains the vertices that are valid to both agents $a_i$ and $a_j$; (2) $X_i = M_i \setminus M_j$ which contains the vertices that is only valid to agent $a_i$ but invalid to agent $a_j$; (3) $X_j = M_j \setminus M_i$ which contains the vertices that is only valid to agent $a_j$ but invalid to agent $a_i$; (4) $X_0 = E \setminus (M_i \cup M_j)$ which contains the vertices that invalid to both $a_i$ and $a_j$. We illustrate the partition in Figure 3.

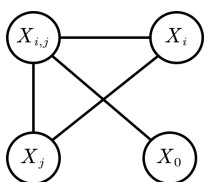

**Figure 3: The illustration of $X_{i,j}, X_i, X_j$ and $X_0$.**

We have the following property for $X_0$. Note that Claim 1 holds for any number of agents.

CLAIM 1. $X_i, X_j$ *and* $X_0$ *are independent sets, i.e., for any edge* $(u, v) \in E$, $|\{u, v\} \cap X_i| \leq 1$, $|\{u, v\} \cap X_j| \leq 1$ *and* $|\{u, v\} \cap X_0| \leq 1$. *Moreover, the vertices in* $X_0$ *are only adjacent to vertices in* $X_{i,j}$.

PROOF. To see $|\{u, v\} \cap X_0| \leq 1$, for the sake of contradiction, suppose there exists an edge $(u, v) \in E$ such that has both $u, v \in X_0$. Since $M_i$ and $M_j$ are vertex covers, then $\{u, v\} \cap M_i \neq \emptyset$ and $\{u, v\} \cap M_j \neq \emptyset$, which contradicts $X_0 = E \setminus (M_i \cup M_j)$. Similarly, if there is an edge $(u, v) \in E$ such that has both $u, v \in X_i = M_i \setminus M_j$, then $\{u, v\} \cap M_j = \emptyset$, which contradicts $M_j$ being a vertex cover. Since $X_i$ and $X_j$ are symmetric, this also holds for $X_j$.

To see the second claim, it suffices to note that for any edge $(u, v) \in E$, if $u \in X_0$, then $u \notin M_i$ and $u \notin M_j$, and thus $v$ must be valid to both agents, i.e., $v \in M_i \cap M_j$. $\square$

LEMMA 4. *For any FSAP instance* $\mathcal{I} = (\mathcal{N}, G = (V, E), \mathbf{c})$ *with* $|\mathcal{N}| = 3$, *there exists a 2-MMS allocation.*

PROOF. For the ease of presentation of this section, we normalize the weight of each vertex such that $\sum_{v \in M_i} w_{i,v} = 1$ for $a_i \in \mathcal{N}$. Thus, by Lemma 2, $\mu_i \leq 1/3$ for $a_i \in \mathcal{N}$. Similar as the main algorithm in the previous section, our idea is to allocate the agents the edges covered by their valid vertices. Depending on whether there are two agents who share many valid vertices, we consider the following three cases.

CASE (1). There exist two agents $a_i, a_j \in \mathcal{N}$ such that

$$\begin{cases} \sum_{v \in X_{i,j}} w_{i,v} \geq \frac{1}{3}, \\ \sum_{v \in X_{i,j}} w_{j,v} \geq \frac{1}{3}. \end{cases}$$

For this case, $X_{i,j}$ contains sufficiently large vertices to the two agents, and we can show that there must be a bundle $S \subseteq X_{i,j}$ that is no more than twice MMS to one of them and at least $\frac{1}{3}$ to the other agent. This can be proved via a bag-filling algorithm. We maintain a bag $B$, which is initially empty. We add vertices in $X_{i,j}$ one by one in an arbitrary order to $B$ until the first time after a vertex $v^*$ is added both agents have weight at least $\frac{1}{3}$ on the bag and we stop. Since $\sum_{v \in X_{i,j}} w_{i,v} \geq \frac{1}{3}$ and $\sum_{v \in X_{i,j}} w_{j,v} \geq \frac{1}{3}$, there must exist such a vertex. Without loss of generality, suppose $a_i$ is the later agent whose weight on $B$ exceeds $\frac{1}{3}$. Thus we have $\sum_{v \in B \setminus \{v\}} w_{i,v} < \frac{1}{3}$, $\sum_{v \in B} w_{i,v} \geq \frac{1}{3}$, and $\sum_{v \in B} w_{j,v} \geq \frac{1}{3}$. Suppose $S \subseteq E$ is the edges that are covered by $B$. Let $A_i = S$, $A_j = E \setminus S$, and $A_{6-i-j} = \emptyset$. Next, we prove that $\mathbf{A} = (A_i, A_j, A_{6-i-j})$ is a 2-MMS allocation, and it suffices to consider $a_i$ and $a_j$.

For agent $a_i$, we have

$$c_i(A_i) \leq \sum_{v \in B} w_{i,v} = \sum_{v \in B \setminus \{v^*\}} w_{i,v} + w_{i,v^*} \leq 2 \cdot \mu,$$

where the first inequality is by Lemma 1, and the second inequality is by Lemma 2 combining $\sum_{v \in B \setminus \{v\}} w_{i,v} < \frac{1}{3}$.

For agent $a_j$, we first note that as $B \subseteq X_{i,j} \subseteq M_j$, $M_j \setminus B$ must be able to cover $E \setminus S$. Since $\sum_{v \in B} w_{j,v} \geq \frac{1}{3}$ and $B \subseteq X_{i,j} \subseteq M_j$, thus we have

$$c_j(A_j) \leq \sum_{v \in M_j \setminus B} \leq \frac{2}{3} \leq 2 \cdot \mu_j,$$

where the two inequalities are again by Lemmas 1 and 2.

Case (2). There exist two agents $a_i$ and $a_j$ that

$$\begin{cases} \sum_{v \in X_{i,j}} w_{i,v} < \dfrac{1}{3}, \\[4mm] \sum_{v \in X_{i,j}} w_{j,v} \ge \dfrac{1}{3}. \end{cases}$$

This case is simple, and can be analyzed in a similar way as Case (1). In fact, $X_{i,j}$ directly satisfies the requirement of the bag $B$ found by the bag-filling algorithm: For $a_i$, $X_{i,j}$ is not too large, and for $a_j$, $X_{i,j}$ is sufficiently large. Denote by $S \subseteq E$ the set of edges covered by the entire $X_{i,j}$. Then let $A_i = S$, $A_j = E \setminus A_i$, and $A_{6-i-j} = \emptyset$. Thus $c_i(A_i) \le \mu_i$ and $c_i(A_j) \le 2\mu_j$.

Case (3). For any two agents $a_i \ne a_j$, we have

$$\begin{cases} \sum_{v \in X_{i,j}} w_{i,v} < \dfrac{1}{3}, \\[4mm] \sum_{v \in X_{i,j}} w_{j,v} < \dfrac{1}{3}. \end{cases}$$

Let $S \subseteq E$ be the set of edges covered by $X_{2,3}$ and $A_2 \leftarrow S$. Let $A_1 = E \setminus A_2$, $A_2 = S$, and $A_3 = \emptyset$. In the following, we show $\mathbf{A} = (A_1, A_2, A_3)$ is a 2-MMS allocation. Again, we only need to check agents $a_1$ and $a_2$.

For agent $a_2$, it is straightforward that

$$c_2(A_2) \le \sum_{v \in X_{2,3}} w_{2,v} \le \frac{1}{3}.$$

For agent $a_j$, we first observe that the vertices in $M_1$ can be divided into two parts $M_1 = (M_1 \cap (M_2 \cup M_3)) \cup (M_1 \setminus (M_2 \cup M_3))$. The vertices in $M_1 \setminus (M_2 \cup M_3)$ are invalid for agent $a_2$ and $a_3$. By Claim 1, any edge $(v, u) \in E \setminus S$ must satisfy that $v \in X_2$ and $u \in X_3$ or the other way round. That is, the vertices in $M_1 \cap (M_2 \cup M_3)$ can cover all the edges $E \setminus S$. Therefore, we have

$$\begin{aligned} c_1(A_1) &\le \sum_{v \in M_1 \cap (M_2 \cup M_3)} w_{1,v} \\ &\le \sum_{v \in M_1 \cap M_2} w_{1,v} + \sum_{v \in M_1 \cap M_3} w_{1,v} \\ &= \sum_{v \in X_{1,2}} w_{1,v} + \sum_{v \in X_{1,3}} w_{1,v} < \frac{1}{3} + \frac{1}{3} < 2\mu_1. \end{aligned}$$

Combing the above three cases, Lemma 4 is proved. □

From the proof of Lemma 4, we can see that if the valid vertices $M_1$, $M_2$ and $M_3$ are given as an oracle, a 2-MMS allocation can be found in polynomial time.

## 4.2 Four Agents

Finally, in this subsection, we show how to find a 2-MMS allocation for any FSAP instance $\mathcal{I} = (\mathcal{N}, G = (V, E), \mathbf{c})$ with 4 agents, i.e., $\mathcal{N} = \{a_1, a_2, a_3, a_4\}$. For every three agents $a_i, a_j, a_k$, we divide the valid vertices of every agent into 4 parts. For agent $a_i$, the valid vertices $M_i$ are divided into four parts as Figure 4 shows: $X_{i,j,k} = M_i \cap M_j \cap M_k$ denotes the set of vertices that are valid to all three agents $a_i$, $a_j$ and $a_k$; $X_{i,j} = (M_i \cap M_j) \setminus M_k$ denotes the set of vertices that are valid to agents $a_i$ and $a_j$ but not to $a_k$;

$X_{i,k} = (M_i \cap M_k) \setminus M_j$ denotes the set of vertices that are valid to agents $a_i$ and $a_k$ but not to $a_j$; $X_{i,0} = M_i \setminus (M_j \cup M_k)$ denotes the set of vertices that are only valid to agent $a_i$ but not to agents $a_j$ and $a_k$. For simplicity, we denote $X_i = X_{i,j,k} \cup X_{i,j} \cup X_{i,k}$ as the vertices in $M_i$ that are valid to at least two agents.

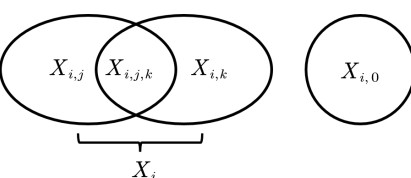

**Figure 4: The division of valid vertices $M_i$ for agent $a_i$.**

By the definition, we have $X_{i,j,k} = X_{j,i,k} = X_{k,i,j}$, $X_{i,j} = X_{j,i}$, $X_{i,k} = X_{k,i}$ and $X_{j,k} = X_{k,j}$. For the ease of presentation, we only use $X_{i,j,k}$, $X_{i,j}$, $X_{i,k}$ and $X_{j,k}$. We have the following property for any three agents $a_i, a_j, a_k$.

**Claim 2.** $X_{i,0}$ is an independent set, i.e., for any edge $(u, v) \in E$, $|\{u, v\} \cap X_{i,0}| \le 1$. Moreover, the vertices in $X_{i,0}$ are only adjacent to vertices in $X_{i,j,k}$.

**Proof.** We first focus on the agents $a_i$ and $a_j$. By Claim 1, it is known that no edge has two endpoints in $X_{i,0} \cup X_{i,k}$ and the edge which has an endpoint in $X_{i,0} \cup X_{i,k}$ must have the other endpoint in $X_{i,j} \cup X_{i,j,k}$. So there is no edge that has two endpoints in $X_{i,0}$.

Then we focus on the agents $a_i$ and $a_k$. By Claim 1, it is known that the edge which has an endpoint in $X_{i,0} \cup X_{i,j}$ must have the other endpoint in $X_{i,k} \cup X_{i,j,k}$. So we have the edge which has an endpoint in $X_{i,0}$ must have the other endpoint in the intersection of $X_{i,j} \cup X_{i,j,k}$ and $X_{i,k} \cup X_{i,j,k}$, which is $X_{i,j,k}$. □

**Lemma 5.** For any FSAP instance $\mathcal{I} = (\mathcal{N}, G = (V, E), \mathbf{c})$ with $|\mathcal{N}| = 4$, there exists a 2-MMS allocation.

**Proof.** We first also normalize the weight of each vertex such that $\sum_{v \in M_i} w_{i,v} = 1$ for $a_i \in \mathcal{N}$. So for any agent in $\mathcal{N}$, her maximin share is no less than $\frac{1}{4}$ by Lemma 2. Here we consider three cases.

Case (1). There exists a group of 3 agents $\{a_i, a_j, a_k\}$ that

$$\sum_{v \in X_l} w_{l,v} \le \frac{1}{2}, \text{ for } l \in \{a_i, a_j\}.$$

Find the set $S$ of edges covered by $X_j$ and allocate the edges in $S$ to agent $a_j$ such that $A_j \leftarrow S$. For the remaining edges in $E$, allocate them to agent $a_i$ $A_i \leftarrow E \setminus S$. So the other two agents $a_k$ and $a_{10-i-j-k}$, there is no edge to allocate $A_k \leftarrow \emptyset$ and $A_{10-i-j-k} \leftarrow \emptyset$.

For agent $a_j$, the cost of her allocation bundle is

$$c_j(A_j) \le \sum_{v \in X_j} w_{j,v} \le \frac{1}{2} \le 2\mu_j.$$

By Claim 2, after allocating $A_j$ to agent $a_j$, any edge covered by $X_{i,0}$ is removed since the edge must be covered by $X_{i,j,k} \subseteq X_j$. $M_i$ can cover $E$, so $X_i$ can cover the remaining edges $E \setminus A_j$. For agent $a_i$, the cost of her allocation bundle is

$$c_i(A_i) \le \sum_{v \in X_i} w_{i,v} \le \frac{1}{2} \le 2\mu_i.$$

CASE (2). There exists a group of agents $\{a_i, a_j, a_k\}$ that

$$
\begin{cases}
\sum_{v \in X_l} w_{l,v} > \frac{1}{2}, \text{ for } l \in \{a_i, a_j\} \\
\sum_{v \in X_k} w_{k,v} \leq \frac{1}{2}.
\end{cases}
$$

Find the set $S$ of edges covered by $X_k$ and allocate agent $a_k$ the edges in $S$: $A_k \leftarrow S$. Let agent $a_i$ divide the vertices in $X_{i,j}$ into two parts $X_{i,j}^1$ and $X_{i,j}^2$ based on the sum weight as equal as possible. Assume that $\sum_{v \in X_{i,j}^1} w_{j,v} \leq \sum_{v \in X_{i,j}^2} w_{j,v}$. Find the set $S'$ of edge covered by $X_{i,j}^1$ in $E \setminus A_i$ and allocate to agent $a_j$: $A_j \leftarrow S'$. For the remaining edges, allocate them to agent $a_i$ such that $A_i \leftarrow E \setminus A_k \setminus A_j$ and $A_{10-i-j-k} = \emptyset$. For agents $a_k$ and $a_j$, we have

$$
c_j(A_j) \leq \sum_{v \in X_{i,j}^1} w_{j,v} \leq \frac{1}{2} \sum_{v \in X_{i,j}} w_{j,v} \leq \frac{1}{2} \leq 2\mu_j,
$$

$$
c_k(A_k) \leq \sum_{v \in X_k} w_{k,v} \leq \frac{1}{2} \leq 2\mu_k.
$$

Based on the discussion of situation 1, the edges covered by $X_{i,0}$ are allocated to agent $a_k$. The edges covered by $(X_{i,k} \cup X_{i,j,k}) \subseteq X_k$ are also allocated to agent $a_k$. So the edges in $A_i$ and $A_j$ can be covered by $X_{i,j}$. For one agent $a_i$'s MMS defining partition $\{B_1, B_2, B_3, B_4\}$, the sum weights of $X_{i,j} \cap (B_1 \cup B_2)$ and $X_{i,j} \cap (B_3 \cup B_4)$ are both less than $2\mu_i$. So the sum weights of $X_{i,j}^1$ and $X_{i,j}^2$ are both less than $2\mu_i$. The edges covered by $X_{i,j}^1$ are allocated to agent $a_j$, so the remaining edges can be covered by $X_{i,j}^2$. For agent $a_i$, we also have

$$
c_i(A_i) \leq \sum_{e \in X_{i,j}^2} w_{i,e} \leq 2\mu_i.
$$

CASE (3). For any three agents $\{a_i, a_j, a_k\}$, we have

$$
\sum_{v \in X_l} w_{l,v} > \frac{1}{2}, \text{ for } l \in \{i, j, k\}.
$$

We first find three agents $a_1$, $a_p$, and $a_q$, and allocate the edges in $E$ to these three agents.

CLAIM 3. *For agent $a_1$, there must be at least two agents $a_p$ and $a_q$ from $a_2, a_3, a_4$ satisfies that $\sum_{v \in M_1 \cap M_l} w_{1,v} \geq \frac{1}{4}$.*

PROOF. For any group of three agents $\{a_1, a_j, a_k\}$ with $a_j \in \{a_2, a_3, a_4\}$, $a_k \in \{a_2, a_3, a_4\}$ and $a_j \neq a_k$, $\sum_{v \in X_1} w_{1,v} > \frac{1}{2}$. In this group, at least one agent between $a_i$ and $a_j$ satisfies that $\sum_{v \in M_1 \cap M_l} w_{1,v} \geq \frac{1}{2}$. Otherwise, we have

$$
\sum_{v \in X_1} \leq \sum_{v \in M_1 \cap M_j} + \sum_{v \in M_1 \cap M_k} < \frac{1}{4} + \frac{1}{4} = \frac{1}{2},
$$

which is a contradiction. So we can find such an agent as $a_p$. Similarly, for the group $\{a_1, a_2, a_3, a_4\} \setminus \{a_p\}$, we can also find such an agent as $a_p$. □

Now we have a group of agents $\{a_1, a_p, a_q\}$. Let the agent $a_{9-p-q}$ which does not belong to the group get no edge such that $A_{9-p-q} \leftarrow \emptyset$. We first keep adding the vertices from $X_{1,p}$ into a bag $B$ until $\sum_{v \in B} w_{1,v} \geq 1/4$ or there is no vertex in $X_{1,p}$. If there is no vertex in $X_{1,p}$ but $\sum_{v \in B} w_{1,v} < \frac{1}{4}$, we keep adding the vertices from $X_{1,p,q}$

into the bag $B$ until $\sum_{v \in B} w_{1,v} \geq \frac{1}{4}$. We have proved there are enough vertices in $X_{1,p} \cup X_{1,p,q}$. Find the set of edges $S$ covered by $B$. If $\sum_{v \in B} w_{p,v} > \frac{1}{2}$, let $A_1 \leftarrow S$, $A_p \leftarrow E \setminus S$ and $A_q \leftarrow \emptyset$. Since for agent $a_1$, the last vertex added to $B$ is valid, the weight of the vertex is less than $\mu_i$. The sum weight of B before adding the last vertex is less than $\frac{1}{4}$. So we have

$$
c_1(A_1) \leq \sum_{v \in B} w_{i,v} \leq \frac{1}{4} + \mu_1 \leq 2\mu_1.
$$

$$
c_p(A_p) \leq 1 - \sum_{v \in B} w_{i,v} < \frac{1}{2} \leq 2\mu_p.
$$

If $\sum_{v \in B} w_{p,v} < \frac{1}{2}$, let $A_p \leftarrow S$ and $c_p(A_p) \leq \sum_{v \in B} w_{p,v} < 2\mu_p$. Next, we start with a new bag $B'$ and keep adding the vertices from $X_{1,q}$ until $\sum_{v \in B'} w_{1,v} \geq \frac{1}{4}$ or there is no vertex in $X_{1,q}$. If $\sum_{v \in B'} w_{1,v} < \frac{1}{4}$, we keep adding the vertices from $X_{1,p,q} - B$ until $\sum_{v \in B'} w_{1,v} \geq \frac{1}{4}$ or there is no vertex in $X_{1,p,q} - B$. Find the set of vertices $S'$ covered by $B'$. If $\sum_{v \in B'} w_{q,v} > \frac{1}{2}$, let $A_1 \leftarrow S'$, $A_q \leftarrow E \setminus (A_1 \cup A_p)$. Like the allocation above, this is also a 2-MMS allocation. If $\sum_{v \in B'} w_{q,v} < \frac{1}{2}$, let $A_q \leftarrow S'$, $A_1 \leftarrow E \setminus (A_p \cup A_q)$. If $\sum_{v \in B'} w_{1,v} < \frac{1}{4}$, it means there is no vertex in $X_{1,p,q} - B$ and $X_{1,p,q} \cap B \neq \emptyset$. So the vertices in $X_1$ are all allocate to $B$ or $B'$ and $\sum_{v \in (B \cup B')} w_{1,v} = \sum_{v \in X_1} > \frac{1}{2}$. Otherwise if $\sum_{v \in B'} w_{1,v} \geq \frac{1}{4}$, it is known that $\sum_{v \in B} w_{1,v} \geq \frac{1}{4}$, so $\sum_{v \in (B \cup B')} w_{1,v} \geq \frac{1}{2}$. We have

$$
c_1(A_1) \leq \sum_{v \in M_1 \setminus (B \cup B')} w_{1,v} = 1 - \sum_{v \in B \cup B'} w_{1,v} \leq \frac{1}{2} \leq 2\mu_1.
$$

$$
c_q(A_q) \leq \sum_{v \in B'} w_{q,v} \leq \frac{1}{2} \leq 2\mu_q.
$$

Combing the above three cases, Lemma 5 is proved. □

## 5  CONCLUSION

This paper introduces the Fair Surveillance Assignment Problem (FSAP), which involves assigning edges of a given graph to a set of agents for surveillance purposes. The agents' cost functions are determined by the weight of the vertex covers within their assigned subgraphs, using their own weight metric. Our primary focus is on achieving MMS fairness in the allocations. For the general FSAP with an arbitrary number of agents, we prove that the optimal approximation ratio lies between 2 and 4.562. Bridging this gap presents an intriguing open problem. Furthermore, we show that the tight approximation ratio bound for scenarios involving no more than four agents is 2. However, it remains uncertain whether these approaches can be extended to the general case.

Our paper uncovers other future research directions as well. Firstly, while our main algorithm computes 4.562-MMS allocations in polynomial time when provided with valid vertices as an oracle, it would be intriguing to enhance this outcome by developing polynomial-time algorithms. Secondly, our paper solely focuses on vertex-cover cost functions, but it would be worthwhile to explore other types of surveillance costs. Thirdly, as general subadditive cost functions do not admit algorithms better than $n$-MMS, it is crucial to identify a class of cost functions, potentially a subset of subadditive functions and a superset of vertex cover functions, that guarantees the existence of constant-approximate MMS allocations.

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
