# OpenReview forum: "Fair Surveillance Assignment Problem"
_ACM.org/TheWebConf/2024/Conference — TheWebConf24_

### Official Review · Reviewer_7r6U · 2023-11-23

**Novelty:** 5
**Technical Quality:** 6

**Review:**

## After rebuttal

I thank the authors for their positive response to the review. After the rebuttal process, my confidence in the paper increased, and I adjusted my score accordingly.

## Summary

The paper works on a surveillance problem, where a set of n agents are assigned edges in a graph for surveillance. The edges are surveilled by covering adjacent vertices, and the paper aims to fairly assign the edges for surveillance to the agents. The agents have vertex-specific costs and suffer the total cost of the minimum vertex cover of the edges they are assigned to surveil. The paper focuses on the well-known maximin fair share property (MMS, as it is defined for chores) and provides three main results: First, no algorithm can guarantee a better than 2-approximate MMS allocation. Then, they present an algorithm that achieves a (nearly) $4.562$-approximation for the specific problem. Finally, they show that an algorithm achieves the optimal approximation of $2$ for at most four agents.

### Evaluation

The paper works on an interesting problem and is relatively well-connected to the conference. The paper provides non-trivial results, and despite some presentation issues, I believe it is a good fit for the conference.

### Pros:

1 - The paper works in an area of active research related to the conference.
2 - The paper provides non-trivial results. The constant approximation mechanism looks impressive compared to what is known in the literature, although this might be due to the particular cost function.
3 - The paper is fairly well-planned and technically sound.

### Cons:

1 - The problem is not as general as communicated in the paper.
2 - the paper seems hastily written, which somehow compromises its quality.

### Detailed comments:

* The cost function for the problem is rather specific, so I don't believe there is any need to denote it in the game instance.
* It is not straightforward how the proposed problem is a generalization of the surveillance problem! Actually, the surveillance problem is never defined.
* Line 133: exists?
* Line 258: $\Pi_n(M)$ --> this is probably $\Pi_n(E)$.
* Line 328: "more notations" --> more notation
* Line 397: "that for which" --> "for which"
* Line 399: "ofG" -> "of G" (missing space)
* Line 411: (Theorem 2 statement): SAP --> FSAP maybe?
* Line 415: On --> One.
* Line 587: there is *a* round.
* Line 598: For the sake of completeness, Theorem 3 should have been proven (by combining Lemmas 4.1 and 4.2).
* Line 606: for $n=2,3$ --> for $n=3,4$.
* Figure 3: The illustration is not explained. What does an edge between two vertices model here?

**Questions:**

1. I cannot get the connection to the Vehicle Routing Problem, as described in the related work. Can you explain this again?

2. The paper does not present a lower bound for Algorithm 1. Do you believe the algorithm has room for improvement? Do you have any algorithm-specific lower bound with an approximation ratio higher than 2?

**Ethics Review Description:**

-

**Reviewer Confidence:**

3: The reviewer is confident but not certain that the evaluation is correct

**Scope:**

4: The work is relevant to the Web and to the track, and is of broad interest to the community

---

### Official Review · Reviewer_xej1 · 2023-11-28

**Novelty:** 5
**Technical Quality:** 4

**Review:**

Summary: This paper studies the Fair Surveillance Assignment Problem where the
 tasks represented by a graph, and the tasks to be assigned correspond to the edges of the graph. The assignment cost of an agent is determined by the minimum vertex cover in the induced subgraph containing the assigned edges (locations). The main focus of the paper is on exploring Maximin Share (MMS) for the assignment. The key findings for MMS fairness include (i) an oracle based algorithm that gives a ~4.562-approximate MMS allocations for heterogeneous agents with arbitrary vertex weights (ii) There doesn’t exist 2-MMS allocations, and (iii) 2-MMS guarantees for n=2,3,4 where n is the number of agents.

Strengths:
1. The  main result of the paper is an oracle-based algorithm providing ~4.562-approximate MMS allocations. This algorithm relies on reducing the edge assignments to vertex assignments after making some noteworthy observations.
2. The paper presents a lower bound example demonstrating the non-existence of 2-MMS allocations for the general FSAP. Additionally, for scenarios where the number of agents is no more than 4, the paper provides oracle-based algorithms that return 2-MMS allocations.
3. This is an interesting model where the allocated item cost of tasks is computed based on a combinatorial structure over tasks, which raises several interesting open questions.

Weaknesses:
1. It is misleading to assert that the primary technical contribution in the paper (Lines: 125-128) lies in computing ~4.562-approximate Maximin Share (MMS) allocations, as this result is contingent on the NP-hard nature of constructing the vertex cover M_i, as explicitly noted in the Remark (Line: 581). This does "not" constitute a polynomial-time algorithm, and the authors should explicitly acknowledge this aspect in the main contributions section [Lines 125-128] and as part of the theorem statement in Theorem 2, thereby characterizing it as an existence result. The same consideration applies to the results in sections 4.1 and 4.2.
2. The paper would benefit from an exploration of polynomial-time algorithms for $\alpha$-MMS allocations, where $\alpha$ is any constant. Such an enhancement would render the results more applicable, especially considering the real-world motivation of the problem in surveillance applications.

**Questions:**

Questions:
1. Are  there any polynomial-time algorithms for constant factor approximations for MMS allocations for arbitrary number of agents or even small number of agents?


Comments:
1. Theorem 3 is a direct corollary of an existing result, hence can be avoided as a Theorem in the main paper.

**Reviewer Confidence:**

3: The reviewer is confident but not certain that the evaluation is correct

**Scope:**

4: The work is relevant to the Web and to the track, and is of broad interest to the community

---

### Official Review · Reviewer_ZN8E · 2023-11-28

**Novelty:** 7
**Technical Quality:** 7

**Review:**

The authors styudy the fair surveillance assignment problem, which is defined as follows: Given a graph $G = (V, E)$ and a set of $N$ agents $a_1, \cdots, a_n$, the task is to allocate the edges of $G$ among these agents. The allocation must satisfy that for each agent $i \in [N]$, the set $A_i$ of edges assigned to $a_i$ should meet two criteria:

1)  $A_i \cap A_j = \emptyset$ for all $i \neq j$, ensuring each edge is uniquely assigned,
2)  $A_1 \cup \cdots \cup A_N = E$, guaranteeing all edges are allocated.

The objective is to find an edge allocation that minimizes the minmax cost across the agents. The cost for each agent $i$, denoted as $c_i(A_i)$, is equivalent to the minimum cost vertex cover for the set $A_i$ according to the weight function $w_i: V \rightarrow \mathbb{R}$.

The main contribution of this paper is to find a solution whose cost is approximately 4.6 times the MMS (MaxiMin Share) cost for each agent, aligning with fairness criteria. The MMS share for an agent $a_i$, defined as $\mu_i$, is the maximin share $\mu_i = \min_{X_1, \cdots, X_n} \max_{j \in [n]} c_i(X_j)$, where $X_1, \cdots, X_n$ represent a partitioning of the edges. The proposed algorithm, which is combinatorial in nature, cleverly leverages the structural properties of the vertex cover. The authors also demonstrate through a hardness example that obtaining an approximation ratio better than 2 with respect to the MMS cost is not feasible, underscoring the solution's effectiveness and limitations.

**Questions:**

The primary limitation, as acknowledged by the authors, relates to the set $M_i$, defined as the union of the minimum cost vertex cover over the minimizer partitions corresponding to each agent's MMS. The current approach exploits oracle access to this object. Could the authors elaborate on the challenges and potential methods for computing the MMS, specifically in terms of hardness and approximability?

Additionally, there appears to be a recurring typographical error in the paper where 'exist' is used instead of 'exit' (see lines 133 and 523, for example). I recommend a thorough review for similar occurrences throughout the manuscript.

Furthermore, I believe the problem would benefit from more robust motivation. One suggestion (which can be ignored as there is no strong tie) is in clustering, which has received considerable attention in literature. It would be beneficial to reference relevant studies, such as those focusing on clustering with min-max fairness or socially fair clustering.

@inproceedings{abbasi2021fair,
  title={Fair clustering via equitable group representations},
  author={Abbasi, Mohsen and Bhaskara, Aditya and Venkatasubramanian, Suresh},
  booktitle={Proceedings of the 2021 ACM conference on fairness, accountability, and transparency},
  pages={504--514},
  year={2021}
}

@inproceedings{ghadiri2021socially,
  title={Socially fair k-means clustering},
  author={Ghadiri, Mehrdad and Samadi, Samira and Vempala, Santosh},
  booktitle={Proceedings of the 2021 ACM Conference on Fairness, Accountability, and Transparency},
  pages={438--448},
  year={2021}
}

@inproceedings{makarychev2021approximation,
  title={Approximation algorithms for socially fair clustering},
  author={Makarychev, Yury and Vakilian, Ali},
  booktitle={Conference on Learning Theory},
  pages={3246--3264},
  year={2021},
  organization={PMLR}
}
@article{goyal2023tight,
  title={Tight fpt approximation for socially fair clustering},
  author={Goyal, Dishant and Jaiswal, Ragesh},
  journal={Information Processing Letters},
  volume={182},
  pages={106383},
  year={2023},
  publisher={Elsevier}
}

@inproceedings{chlamtavc2022approximating,
  title={Approximating Fair Clustering with Cascaded Norm Objectives},
  author={Chlamt{\'a}{\v{c}}, Eden and Makarychev, Yury and Vakilian, Ali},
  booktitle={Proceedings of the 2022 Annual ACM-SIAM Symposium on Discrete Algorithms (SODA)},
  pages={2664--2683},
  year={2022},
  organization={Society for Industrial and Applied Mathematics}
}

**Reviewer Confidence:**

4: The reviewer is certain that the evaluation is correct and very familiar with the relevant literature

**Scope:**

3: The work is somewhat relevant to the Web and to the track, and is of narrow interest to a sub-community

---

### Decision · Program_Chairs · 2024-01-22

**Decision:**

Accept

**Comment:**

I will spare the authors another summary of their work, given the detailed reviews prepared by several of the PC members -- I thank them for their efforts in evaluating this work, both in the original reviews and in the subsequent rebuttal period; all 3 are excellent.

 Given the strong support for acceptance, I think the decision should be clear for this paper.